# FedADM: Adaptive Federated Learning via Dissimilarity Measure

## Abstract

In federated learning, there are two critical challenges: 1) the data on distributed learners is heterogeneous; and 2) communication resources within the network are limited. In this work, we propose a framework, Federated Adaptive Dissimilarity Measure (FedADM), which can be regarded as an adaptively enhanced version of the Federated Proximal (FedProx) algorithm. This adaptiveness is primarily manifested in two aspects: (i) how it adaptively adjusts the proximity between the local models on different learners and the global model; and (ii) how it adaptively aggregates local model parameters. Building on the FedProx model, FedADM incorporates the concept of the Lagrangian multiplier to control the proximal coefficients of different learners, using "*parameter dissimilarity*" to address data heterogeneity. It explicitly captures the essence of using "*loss dissimilarity*" to adaptively adjust the aggregation frequency on distributed learners, thereby reducing communication overhead. Theoretically, we provide the performance upper bounds and convergence analysis of our proposed FedADM. Experiment results demonstrate that FedADM allows for higher accuracy and lower communication overhead compared to the baselines across a suite of realistic datasets.

## 1 Introduction

Data privacy and security are of paramount importance, especially in highly sensitive sectors such as healthcare, finance, and smart manufacturing. Typically, data in different institutions or departments is stored independently, making it challenging to effectively integrate and utilize this dispersed data architecture. Federated learning technology (Kairouz et al., 2021; Chen et al., 2024; Wang et al., 2024b) offers a potent solution to this "data island" issue, enabling the collaborative use of multi-source data across institutions for distributed model training. This approach allows for the resolution without the direct exchange of any sensitive information, thus ensuring the security of the data (Yang et al., 2023; Zhang et al., 2023; Hu et al., 2023).

Federated learning represents a promising method of distributed machine learning training, particularly showing distinct advantages over other traditional distributed optimization methods in heterogeneous data settings (Yang et al., 2019; Chen et al., 2021). The Federated Proximal (FedProx) algorithm is a classic federated learning approach tailored for heterogeneous data distributions (Li et al., 2020). It incorporates a proximal term in local model training, which helps the local models converge towards the global aggregated model, thereby accelerating learning in heterogeneous data and promoting model convergence. However, it lacks in-depth exploration and utilization of heterogeneous data and does not adequately consider resource consumption. Although there has been significant work in federated learning optimizing participant selection (Cho et al., 2022; Tang et al., 2022), local update frequency (Singhal et al., 2021; Ruan & Joe-Wong, 2022), and aggregation count (Pillutla et al., 2022; Zhang et al., 2023; Li et al., 2023; Wang et al., 2024b; Lee et al., 2023) to reduce overheads, achieving global optimum remains challenging. A pivotal question of federated learning regarding the data heterogeneity and limited communication resources that emerge in our research is:

**Question 1** *How can we deeply excavate and precisely harness the heterogeneity inherent in data to guide the local updates and global aggregation processes in federated learning?*

In this work, we introduce the Federated Adaptive Dissimilarity Measure (FedADM), a federated learning approach that addresses this question. The effective utilization of heterogeneous information directly impacts the performance and speed of model learning. Heterogeneity in distributed data can be measured through network parameters, loss functions, and gradient information. *The FedADM approach firstly utilizes a variety of heterogeneity metrics, including parameter dissimilarity and loss dissimilarity, to improve the training efficiency and accuracy of the model.* Our contributions to this work are as follows:

- *Parameter Dissimilarity for Local Network*: We construct a proximal term in the local loss functions by utilizing the *parameters dissimilarity* between local and global aggregation. Note that this approach dynamic controls the influence of parameter dissimilarity on the local loss function by adjusting the proximal coefficient, thus promoting the convergence of the local model.

- *Loss Dissimilarity for Surrogate Function*: The framework utilizes local *loss dissimilarity* to construct a surrogate for the optimization objective function. This effectively captures the variations in local models and theoretically achieves a suboptimal solution for the number of local updates. This strategy aids in optimizing the frequency of local updates and global aggregations.

- *Experiment Results*: Extensive experiments verify the effectiveness and convergence of the FedADM under limited communication resources using three real datasets, four cases with heterogeneous data, various neural network models, and different system configurations.

## 2 RELATED WORK

Security has significantly advanced the development of federated learning technologies within the field of distributed optimization (Kim et al., 2023; Ye et al., 2023; Tang et al., 2024). Generally, assumed that the data points are non-independent and non-identically distributed (non-IID) (Pillutla et al., 2022; Liao et al., 2023). These increase the difficulty of federated learning.

**Data Heterogeneity** FedProx, a seminal work, introduced a proximal term to facilitate training collaboration across heterogeneous data sources (Li et al., 2020). Building on this, reference (Wu et al., 2023) defined the local objective function to incorporate the momentum-based variance-reduced technique. (Wu et al., 2024) modeled federated learning as non-convex minimax optimization problems. Moreover, (Pathak & Wainwright, 2020) developed the FedSplit method employing operator splitting, and (Zhao et al., 2023) decomposed an upper bound of the objective into a bias term and a variance term to achieve a trade-off between heterogeneity and aggregation. The above works focus on reconstructing the objective function, which lacks a deep exploration of heterogeneity. Our work analyzes and utilizes dissimilarity in parameters and losses and then reconstructs objective guiding model optimization to achieve more accuracy and robustness.

**Aggregation** Automated aggregation control significantly enhanced the efficiency of federated learning and reduced communication overhead, as demonstrated through methods like simple weighted aggregation (Li et al., 2023), two-stage clustering aggregation (Zhou et al., 2024), cross-round aggregation (Wang et al., 2024a), and layer-wise aggregation (Lee et al., 2023; Chan et al., 2023). Although existing aggregation methods improved the communication efficiency of federated learning (Nguyen et al., 2022; Wang et al., 2021; Chu et al., 2022; An et al., 2023; Chen et al., 2022), they often did not take into account dynamically constrained communication resources.

**Convergence Guarantee** The theoretical guarantees for the convergence of models in federated learning had been extensively studied (Mitra et al., 2021; Koloskova et al., 2022; Charles & Konečnỳ, 2021; Zhang et al., 2022; Gao et al., 2021). These works included convergence guarantees for non-convex federated optimization (Yuan & Li, 2022) and asynchronous federated learning (Bornstein et al., 2022). Furthermore, the work by (Nguyen et al., 2020) designed a fast-convergent mechanism and theoretically verified the improvement of a lower bound for local models. To simplify optimization problems, we use surrogate functions to prove the models' convergence.

## 3 FEDRATED OPTIMIZATION: METHODS

A federated learning model is considered, operating within a constrained resource budget $R$. This model consists of a central controller and $n$ local learners. Each learner maintains its own network parameters and performs *local updates* during the training process. After every $\tau$ local update, learners transmit their model updates to the central controller. Upon receiving these updates, the central controller performs a *global aggregation* and redistributes the results back to the local learners. Define $T$ as the total number of iterations conducted by each learner, with each iteration denoted by $t$, where $t \in \{1, 2, \ldots, T\}$. Let $K$ represent the number of aggregations, calculated as $T/\tau$, with each aggregation indexed by $k$, where $k \in \{1, 2, \ldots, K\}$. Both local updates and global aggregations during the federated learning process consume resources, which may include transmission bandwidth, storage capacity, and computational power. Consider $M$ distinct types of resources, each type labeled as $m$, where $m \in \{1, 2, \ldots, M\}$. The variables $c_m$ and $b_m$ denote the resource consumption per local update and per global aggregation, respectively, for the $m$-th type of resource. $R_m$ represents the total available budget for resource type $m$. The primary objective is to optimize the frequency of local updates and global aggregations by minimizing the loss function of the central controller, all while adhering to the constraints imposed by the limited resources. This optimization problem can be formally expressed as follows:

$$
\begin{aligned}
\min_{\tau, K} \quad & F(\mathbf{w}) \\
\text{s.t.} \quad & Tc_m + Kb_m \leq R_m, \forall m \in \{1, 2, ..., M\} \\
& T = K\tau.
\end{aligned}
\tag{1}
$$

### 3.1 FEDERATED PROXIMAL (FEDPROX)

The datasets on the learners are often non-IID, resulting in varying labels and data quantities among the learners. This non-IID distribution can cause model performance variability, convergence difficulties, and overfitting issues. To address these challenges, FedProx introduces a proximal term during local training. The FedProx method trains a model that minimizes global loss, ensuring that local model parameters do not deviate excessively from the global model parameters. The objective function of FedProx for local learner $i$ with a proximal term is described as follows:

$$
F_i^p(\mathbf{w}_i) = F_i(\mathbf{w}_i) + \frac{\mu}{2} \|\mathbf{w}_i - \mathbf{w}_{k,global}\|^2,
\tag{2}
$$

where $\mathbf{w}_i$ are the parameters of the $i$-th learner, $i \in \{1, 2, \ldots, n\}$, and $\mathbf{w}_{k,\text{global}}$ are the global parameters from the $k$-th aggregation, the proximal term coefficient is represented by $\mu$. This term adjusts the closeness of local updates to the initialized global model. The global loss function, relevant to the $i$-th learner's dataset, is denoted by

$$
F^p(\mathbf{w}) = \sum_{i=1}^{n} F_i^p(\mathbf{w}),
\tag{3}
$$

where $\mathbf{w}$ represents the central controller's model parameters, obtained through global aggregation. Specifically, $\mathbf{w} = \frac{1}{D} \sum_{i=1}^{n} D_i \mathbf{w}_i$ denotes the weighted average of the local model parameters from all learners participating in the aggregation, where $D_i$ is the number of samples at the $i$-th learner. Here, $D$ is the sum of the number of samples across all learners, given by $D = \sum_{i=1}^{n} D_i$. The optimal model parameters at the $k$-th aggregation are

$$
\mathbf{w}_{k,global} \stackrel{\Delta}{=} \underset{\mathbf{w} \in \{\mathbf{w}(k\tau): k=1,2,\ldots,K\}}{\arg\min} F^p(\mathbf{w}).
\tag{4}
$$

### 3.2 DEFINITIONS IN FEDADM

**Definition 1 (Bounded Parameter Dissimilarity)** *An upper bound of the parameter dissimilarity between the parameters of the $i$-th learner $\mathbf{w}_i$, and the global parameters of the $k$-th aggregation $\mathbf{w}_{k,global}$, is given by*

$$
\|\mathbf{w}_i - \mathbf{w}_{k,global}\| \leq \xi,
\tag{5}
$$

*where $\xi$ is a predefined parameter deviation tolerance.*

Since the data distribution varies across different learners, setting dynamic control parameters for the proximal term allows for more precise control over its importance in the training process. Specifically, the objective function for local learners with adaptive proximal terms is described as follows:

$$F_i^p\left(\mathbf{w}_i, \mu_{i,k}\right) = F_i\left(\mathbf{w}_i\right) + \frac{\mu_{i,k}}{2}\left\|\mathbf{w}_i - \mathbf{w}_{k,global}\right\|^2, \tag{6}$$

where $\mu_{i,k}$ denotes the proximal term coefficient for the $i$-th learner at the $k$-th aggregation. The parameters of the $i$-th learner are $\mathbf{w}_i$, and the global parameters of the $k$-th aggregation are $\mathbf{w}_{k,\text{global}}$. It is easy to see that equation 6 can be regarded as the Lagrangian function of the local objective function, which satisfies the bounded parameter dissimilarity condition stated in Definition 1. The proximal term coefficient $\mu_{i,k}$ acts as a regulator of sensitivity constraints, adaptively adjusting $\mu_{i,k}$ by utilizing the *parameter dissimilarity* between $\mathbf{w}_i$ and $\mathbf{w}_{k,\text{global}}$. With each local gradient aggregation, a new $\mathbf{w}_{k,\text{global}}$ is obtained, and thus, the proximal coefficient is updated by

$$\mu_{i,k} = \mu_{i,k-1} + \alpha(\left\|\mathbf{w}_i - \mathbf{w}_{k,global}\right\| - \xi), \tag{7}$$

where $\alpha$ represents the learning rate that regulates the update speed of the proximal coefficients. This update method enhances the regularization effect by increasing the proximal term coefficient when the local model parameters significantly deviate from the global model, thereby compelling the local model to align more closely with the global model. Conversely, if the local model parameters are close to the global model, the proximal coefficient is reduced.

The local training parameters are updated using the gradient descent method that is given by:

$$\mathbf{w}_i(t) = \mathbf{w}_i(t-1) - \eta\left(\nabla F_i\left(\mathbf{w}_i(t-1)\right) + \mu_{i,k}\left\|\mathbf{w}_i(t-1) - \mathbf{w}_{k,global})\right\|\right), \tag{8}$$

where $\eta$ is a given learning rate. To effectively manage the data heterogeneity across various learners, it is crucial to analyze the relationship between the local and global loss functions:

**Definition 2 (Local Loss Dissimilarity)** *The local loss dissimilarity captures the heterogeneity of the local network, which is modeled by:*

$$\frac{1}{n}\sum_{i=1}^n\left\|F_i\left(\mathbf{w}_{k,global}\right)\right\|^2 \le B_k^2\|F\left(\mathbf{w}_{k,global}\right)\|^2 + H_k^2, \tag{9}$$

*which fits the relationship between the behavior of local loss and the global loss, scaled by $B_k$ and adjusted by a constant $H_k$ at the $k$-th aggregation.*

When $B_k$ approaches 1 and $H_k$ nears 0, it indicates that the gradient at each learner closely aligns with the global gradient. The $B_k$ and $H_k$ are updated by

$$B_k = \sqrt{\frac{\sum_{i=1}^n\left\|F_i(\mathbf{w}_{k,global})\right\|^2}{n\cdot\left\|F(\mathbf{w}_{k,global})\right\|^2}}, \tag{10}$$

$$H_k = \sqrt{\max\left(0, \frac{1}{n}\sum_{i=1}^n\left\|F_i(\mathbf{w}_{k,global})\right\|^2 - B_k^2\left\|F(\mathbf{w}_{k,global})\right\|^2\right)}. \tag{11}$$

## 4 FedADM: Theoretical Analysis

The specific expression of the variables $\tau$ and $K$ in the objective function equation 1 is analytically challenging for two main reasons: (i) it depends on the convergence characteristics of the gradient; (ii) resource consumption dynamically changes. The sketch of the theoretical analysis is as follows: First, analyze the upper bound of convergence and use this boundary to approximate the solution to equation 1. Based on the local *loss dissimilarity*, a linear search method is used to optimize $\tau$ and $K$, obtaining the asymptotically optimal solution for equation 1.

## 4.1 CONVERGENCE ANALYSIS

The convergence of Algorithm 1 is verified, and an upper bound is obtained between the loss function $F(\mathbf{w}_{k,\text{global}})$ with aggregation parameters and the loss function $F(\mathbf{w}^*)$ with optimal parameters. The convergence analysis of Algorithm 1 comprises two steps: (i) measuring the gap in parameters between distributed gradient descent and centralized gradient descent at the $k\tau$-th iteration (note that federated learning has not yet undergone global aggregation at the $k\tau$-th iteration when computing the gap); (ii) combining the gap identified in the first step with the convergence upper bound of centralized gradient descent to derive the upper bound of convergence for $\mathbf{w}$.

To facilitate the convergence analysis, we also consider a centralized training framework that involves only a global neural network with no local learners. All information is observable, and network parameters are updated using the centralized gradient descent method. To ensure a fair comparison between distributed federated learning and centralized learning, it is crucial to maintain consistency in the loss functions used in both approaches. In centralized training, this involves leveraging the global aggregation parameters $\mathbf{w}_{k,\text{global}}$ from federated learning to construct a proximal term. Similarly, the centralized network incorporates a proximal term at the $k$-th interval that is given by:

$$\mathbf{v}_k(t) = \mathbf{v}_k(t-1) - \eta\left(\nabla F\left(\mathbf{v}_k(t-1)\right) + \bar{\mu}_k\left(\mathbf{v}_k(t-1) - \mathbf{w}_{k,global}\right)\right), \tag{12}$$

where $\bar{\mu}_k = \frac{1}{n}\sum_{i=1}^{n}\mu_{i,k}$ and $t \in [(k-1)\tau, k\tau]$ for a given $k$, $k = 1, 2, ..., K$.

We establish the following assumptions regarding the loss function used in local training.

**Assumption 1** *For any learner $i$, we have:*

*1) $F_i^p(\mathbf{w}_i, \mu_{i,k})$ is convex.*

*2) $F_i^p(\mathbf{w}_i, \mu_{i,k})$ is $(\rho_i + \mu_{i,k}\xi)$-Lipschitz, which means that there exists a constant $\rho_i > 0$ such that for any $\mathbf{w}_i, \mathbf{w}_i'$, the following inequality holds:*

$$\|F_i^p(\mathbf{w}_i, \mu_{i,k}) - F_i^p(\mathbf{w}_i', \mu_{i,k})\| \le (\rho_i + \mu_{i,k}\xi)\|\mathbf{w}_i - \mathbf{w}_i'\|. \tag{13}$$

*3) $F_i^p(\mathbf{w}_i, \mu_{i,k})$ is $(\beta_i + \mu_{i,k})$-Smooth, which means that there exists a constant $\beta_i > 0$ such that for any $\mathbf{w}_i, \mathbf{w}_i'$, the following inequality holds:*

$$\|\nabla F_i^p(\mathbf{w}_i, \mu_{i,k}) - \nabla F_i^p(\mathbf{w}_i', \mu_{i,k})\| \le (\beta_i + \bar{\mu}_k)\|\mathbf{w}_i - \mathbf{w}_i'\|. \tag{14}$$

Based on Assumption 1, we can derive the properties of the central controller's loss function: $F^p(\mathbf{w})$ is convex, $(\rho + \mu_{i,k}\xi)$-Lipschitz, $(\beta + \mu_{i,k})$-Smooth (Wang et al., 2019), where $\rho = \frac{1}{D}\sum_{i=1}^{n}D_i\rho_i$ and $\beta = \frac{1}{D}\sum_{i=1}^{n}D_i\beta_i$. Moreover, it describes the differences between the gradients of the local learner loss and the global loss. This divergence measures how the data is distributed across different learners.

**Definition 3 (Gradient Divergence)** *An upper bound of the differences is:*

$$\|\nabla F_i^p(\mathbf{w}_i, \mu_{i,k}) - \nabla F^p(\mathbf{w}, \bar{\mu}_k)\| \le \delta_i \tag{15}$$

*where $\nabla F^p(\mathbf{w}, \bar{\mu}_k) = \frac{1}{n}\sum_{i=1}^{n}\nabla F_i^p(\mathbf{w}_i, \mu_{i,k})$ and $\delta = \frac{1}{D}\sum_{i=1}^{n}D_i\delta_i$.*

Using the aforementioned assumptions and definitions, we first find the gap between the parameters of distributed gradient descent $\mathbf{w}(t)$ and centralized gradient descent $\mathbf{v}_k(t)$ as follows:

**Lemma 4.1** *For any interval $k$, and any iteration $t \in [(k-1)\tau, k\tau]$*

$$\|\mathbf{w}(t) - \mathbf{v}_k(t)\| \le h(t - (k-1)\tau), \tag{16}$$

*where*

$$h(x) = \delta\sum_i\left(\beta_\mu\frac{(\eta(\beta + \mu_{i,k}) + 1)^x - 1}{n(\beta + \mu_{i,k})} - \frac{\eta x\beta_\mu}{n}\right), \tag{17}$$

*and $\beta_\mu = \frac{\beta + \bar{\mu}_k}{\beta + \mu_{i,k}}$, $x$ is a non-negative integer.*

From Lemma 4.1, it is evident that $\mu$ significantly influences the upper bound of parameter convergence. Given that the data distribution is non-IID, we frequently observe that $\bar{\mu}_k \neq \mu_{i,k}$, leading to more intricate behavior in the functional response. Consequently, selecting an optimal $\mu_{i,k}$ is crucial for enhancing convergence rates and improving accuracy. A closer analysis of the function $h(x)$ reveals the presence of $x$ in the exponent, which indicates that larger values of $x$, corresponding to a greater number of local training iterations, result in substantially increased function values. This suggests that an increase in local training iterations amplifies the disparity between the model parameters in federated learning compared to those in centralized learning. The proof can be found in Appendix A. Based on the results in Lemma 4.1, we further derive the upper bound on the difference between the two loss functions $F^p(\mathbf{w}(T), \bar{\mu}_K) - F^p(\mathbf{w}^*, \bar{\mu}_K)$:

**Lemma 4.2** *Under the following conditions:*

*1)* $\eta \leq \frac{1}{\beta + \max \mu_{i,k}}$

*2)* $\eta \varphi - \frac{(\rho + \max \mu_{i,K})}{\tau \xi^2} > 0$

*3) For any $k$, we have* $F^p(\mathbf{v}_k(k\tau), \bar{\mu}_k) - F^p(\mathbf{w}^*, \bar{\mu}_K) \geq \varepsilon$

*4)* $F^p(\mathbf{w}(T), \bar{\mu}_K) - F^p(\mathbf{w}^*, \bar{\mu}_K) \geq \varepsilon$

*where $\varepsilon > 0$ and $\varphi = \left(1 - \frac{(\beta + \bar{\mu}_k)\eta}{2}\right) \min_k \frac{1}{\|\mathbf{v}_k((k-1)\tau) - \mathbf{w}^*\|^2}$, $\bar{\mu} = \frac{1}{K} \sum_{k=1}^{K} \mu_k$, then the upper bound of the objective function as follows:*

$$F^p(\mathbf{w}(T), \bar{\mu}_K) - F^p(\mathbf{w}^*, \bar{\mu}_K) \leq \frac{1}{T\left(\eta\varphi - \frac{\rho h(\tau)}{\tau \varepsilon^2}\right) - \frac{\bar{\mu}_K \xi h(\tau)}{\varepsilon^2} - \frac{\xi^2}{2\varepsilon^2} \sum_{k=1}^{K-1} (\bar{\mu}_{k+1} + \bar{\mu}_k)}. \quad (18)$$

Lemma 4.2 establishes an upper bound on the gap between the loss functions at the parameters set to $\mathbf{w}(T)$ and the optimal solution $\mathbf{w}^*$. This relationship is also influenced by $\mu_{i,k}$ and $\xi$. Notably, this upper bound is inversely proportional to the number of training iterations, $T$. Thus, as training progresses, the objective function approaches convergence. A detailed proof is provided in the Appendix B. Building on these insights, the subsequent Theorem 1 elaborates on the further derivation of these bounds.

**Theorem 1 (Upper Bound of Surrogate Function)** *When $\eta \leq \frac{1}{\beta + \max \mu_{i,k}}$, we have*

$$F^p(\mathbf{w}_{k,global}, \bar{\mu}_K) - F^p(\mathbf{w}^*, \bar{\mu}_K) \leq \frac{1}{4\eta\varphi T} + h'(\tau) + (\rho + \bar{\mu}_K \xi) \rho h(\tau), \quad (19)$$

*where*

$$h'(\tau) = \sqrt{\frac{2T\rho h(\tau) + 2\tau \bar{\mu}_K \xi h(\tau) + \tau(\bar{\mu}_K - \bar{\mu}_1)}{\eta\varphi T^2}}. \quad (20)$$

In Theorem 1, the optimal gap $F^p(\mathbf{w}_{k,global}, \bar{\mu}_K) - F^p(\mathbf{w}^*, \bar{\mu}_K)$ is related to $h(\tau)$, where $\delta$ within $h(\tau)$ incorporates information about the data distribution across different learners. With a fixed total number of iterations $T$, the optimal gap increases as $\tau$ and $\delta$ increase. Given $\tau$ and $\delta$, as iterations $T$ grow larger, the optimal gap decreases. Notably, when $\tau = 0$ (i.e., gradient aggregation occurs after every local update), both $h(\tau)$ and $h'(\tau)$ tend towards zero, and as $T$ approaches infinity, the optimal gap tends towards zero. This indicates that the algorithm's solution becomes closer to the optimal solution. However, due to limited resources and typically $\tau > 1$, as $T$ tends towards infinity, convergence is only possible to a non-zero optimal gap. The detailed proof can be found in Appendix C.

## 4.2 FedADM Algorithm

In this subsection, a surrogate function for the loss function equation 1 of the central controller is constructed. The loss function equation 1 of the central controller is challenging due to the incorporation of local gradient updates and the real-time fluctuations in resource consumption. Therefore,

the upper bound $F^p(\mathbf{w}_{k,\text{global}}, \bar{\mu}_K) - F^p(\mathbf{w}^*, \bar{\mu}_K)$ from Theorem 1 is utilized to approximate loss function equation 1, the optimal loss value $F^p(\mathbf{w}^*, \bar{\mu}_K)$ is a constant, the minimization of loss $F^p(\mathbf{w}_{k,\text{global}}, \bar{\mu}_K)$ is equal to minimize $F^p(\mathbf{w}_{k,\text{global}}, \bar{\mu}_K) - F^p(\mathbf{w}^*, \bar{\mu}_K)$. The $K$ in equation 1 is satisfied with:

$$K \leq \frac{R_m}{c_m \tau + b_m}, \forall m \in \{1, 2, \ldots, M\}. \tag{21}$$

We aim to minimize the upper bound of the loss function's gap, thereby finding the optimal value of $\tau$. Based on equation 21 and $T = K\tau$, $T$ is replaced with $\max_m \frac{c_m \tau + b_m}{R_m \tau}$, the upper bound is redefined as:

$$G(\tau) = \frac{\max_m \frac{c_m \tau + b_m}{R_m \tau}}{4\eta\varphi} + h'(\tau) + \left(\rho + \bar{\mu}_{T/\tau}\xi\right)\rho h(\tau), \tag{22}$$

where the given $\eta \leq \frac{1}{\beta}$. The surrogate function of the loss function equation 1 is rewritten as:

$$\begin{aligned} \min_{\tau, K \in \{1,2,3,\ldots\}} \quad & G(\tau) \\ \text{s.t.} \quad & K \leq \frac{R_m}{c_m \tau + b_m}, \forall m \in \{1, 2, \ldots, M\} \\ & T = K\tau, \end{aligned} \tag{23}$$

from which the approximately optimal $\tau^*$, $K^*$, and $T^*$ are

$$\tau^* = \arg\min_\tau G(\tau), \quad K^* = \min_m \frac{R_m}{c_m \tau + b_m}, \quad T^* = \min_m \frac{R_m}{c_m \tau + b_m}\tau^*. \tag{24}$$

During each aggregation, the estimated upper bound in equation 22 is used to replace the challenging primal problem equation 1. A linear search method is then employed to find the value of $\tau^*$ that minimizes equation 23, which is then used to determine the number of local update times for the next interval.

The detailed process of the FedADM algorithm with distributed gradient descent is as follows: We first initialize the parameters. **Local Updates**: Within each global aggregation $k$, each learner $i$ performs $\tau$ iterations of local updates. Each learner receives the latest global model parameters $\mathbf{w}_{k,\text{global}}$ and computes $\mathbf{w}_i(t)$ updates via equation 6, along with $\mu_{i,k}$ via equation 7. The Lipschitz constant is updated applying $\rho_i = \|F_i^p(\mathbf{w}_i, \mu_{i,k}) - F_i^p(\mathbf{w}'_i, \mu_{i,k})\| / \|\mathbf{w}_i - \mathbf{w}'_i\| - \mu_{i,k}\xi$, and the smoothness constant is given by $\beta_i = \|\nabla F_i^p(\mathbf{w}_i, \mu_{i,k}) - \nabla F_i^p(\mathbf{w}'_i, \mu_{i,k})\| / \|\mathbf{w}_i - \mathbf{w}'_i\| - \bar{\mu}_k$. Updates $F_i^p(\mathbf{w}_i, \mu_{i,k})$ and $\nabla F_i^p(\mathbf{w}_i, \mu_{i,k})$ are then sent back to the central controller. **Global Aggregation**: The central controller aggregates updates from all local learners using a weighted average to update the global model parameters $\mathbf{w}(t)$. The loss of the central controller $F^p(\mathbf{w}(t))$ is computed by equation 3. If $F^p(\mathbf{w}(t)) < F^p(\mathbf{w}_{k,\text{global}})$ is satisfied, the parameters $\mathbf{w}_{k,\text{global}}$ are set to $\mathbf{w}(t)$. **Parameter Estimation**: After updating the global model, parameters related to loss such as $\hat{\rho}$, $\hat{\beta}$, and $\hat{\delta}$ are estimated by $\hat{\rho} = \frac{1}{D}\sum_{i=1}^n D_i\rho_i$, $\hat{\beta} = \frac{1}{D}\sum_{i=1}^n D_i\beta_i$, $\hat{\delta} = \frac{1}{D}\sum_{i=1}^n D_i\delta_i$ and $\nabla F^p(\mathbf{w}, \bar{\mu}_k) = \frac{1}{n}\sum_{i=1}^n \nabla F_i^p(\mathbf{w}, \mu_{i,k})$. The parameters of resource consumption $\hat{c}_m$ and $\hat{b}_m$ are also estimated. The parameters $B_k$ and $H_k$ in local loss dissimilarity are obtained from equation 10 and equation 11. The remaining parameter $\varphi$ is regarded as a given control parameter because it includes the unknown $\mathbf{w}^*$. **Compute $\tau^*$ and $K^*$**: By utilizing *loss dissimilarity*, we construct the surrogate function equation 23 to replace the primal optimization problem equation 1. A centralized controller then uses line search to compute $\tau^*$, subsequently calculating $K^*$ and $T^*$. If a STOP flag is met (e.g., the number of iterations has reached the predefined maximum $T$ or resource consumption exceeds limit $R_m$ or $F^p(\mathbf{w}_{k,\text{global}}, \bar{\mu}_K) - F^p(\mathbf{w}^*, \bar{\mu}_K)$ tends to zero), the process progresses to termination; otherwise, it continues with further iterations.

As demonstrated in Algorithm 1, it exhibits markedly low computational complexity. For each global aggregation, the central controller collects parameters from $n$ participating learners, encompassing $M$ types of resources. The number of steps required for local updates, obtained through line search, does not exceed $T^s_{\max}$. The total count of global aggregations is denoted by $K$, and the computational complexity of this global aggregation phase is $O(K(nM + T^s_{\max}))$. Regarding the local gradient updates, these are executed $T$ times in total. Additionally, each local learner processes $M$ resource types during each aggregation phase, leading to enhanced local computations. The computational complexity for local updates across all learners amounts to $O(T + KM)$. Consequently, the overall computational complexity of Algorithm 1 is computed as $O(K(nM + T^s_{\max} + M) + T)$, balancing both global and local computational demands efficiently.

---

**Algorithm 1** FedADM Algorithm for Federated Learning

---

1: Initialize $\mathbf{w}(0), \mathbf{w}_{k,global} \leftarrow \mathbf{w}(0), \tau^* \leftarrow 1, t \leftarrow 1, k \leftarrow 1$.
2: Initialize $\mathbf{w}_i(0)$ to all learners $i \in \{1, 2, \ldots, n\}$.
3: **for** $k = 1$ to $K$ **do**
4:  **for** $t = 1$ to $\tau^*$ **do**
5:   **for** each learner $i$ **in parallel do**
6:    Receive current global model parameters $\mathbf{w}_{k,global}$.
7:    Compute local updates with *parameter dissimilarity*:

$$\min_{\mathbf{w}_i} \left\{ F_i^p \left(\mathbf{w}_i, \mu_{i,k}\right) = F_i \left(\mathbf{w}_i\right) + \frac{\mu_{i,k}}{2} \left\| \mathbf{w}_i - \mathbf{w}_{k,global} \right\|^2 \right\}.$$

8:    Send the update $\Delta \mathbf{w}_i(t) = \mathbf{w}_i(t) - \mathbf{w}_i(t-1)$ to the server.
9:    Send the update $\mu_{i,k}, \hat{\rho}_i, \hat{\beta}_i, F_i^p(\mathbf{w}_i, \mu_{i,k}), \nabla F_i^p(\mathbf{w}_i, \mu_{i,k})$ to the server.
10:   **end for**
11:   Aggregate updates at the server:

$$\mathbf{w}(t) = \mathbf{w}(t-1) + \frac{1}{n} \sum_{i=1}^{n} \Delta \mathbf{w}_i(t).$$

12:  **end for**
13:  Compute $F^p(\mathbf{w}(t))$ according to equation 3.
14:  $\mathbf{w}_{k,global} \leftarrow \mathbf{w}(t)$ if $F^p(\mathbf{w}(t)) < F^p(\mathbf{w}_{k,global})$.
15:  Estimate $\hat{\rho}, \hat{\beta}, \hat{\delta}, B_k, H_k$.
16:  Estimate resource consumption $\hat{c}_m$ and $\hat{b}_m$.
17:  Check for STOP flag: If the STOP flag is true, stop.
18:  Compute $\tau^*$ in equation 23 with *loss dissimilarity*.
19:  Compute $K^*$, and $T^*$.
20:  $k \leftarrow k + 1$.
21: **end for**

---

## 5 SIMULATION RESULTS

### 5.1 EXPERIMENTAL SETUP

**Datasets and Models**: We conduct experiments using three popular datasets: MNIST (LeCun et al., 1998), Fashion-MNIST (Xiao et al., 2017), and CIFAR-10 (Krizhevsky et al., 2009). For the MNIST and Fashion-MNIST datasets, a simple Support Vector Machine (SVM) Cortes (1995) serves as the backbone for training and testing, while for CIFAR-10, we use a Convolutional Neural Network (CNN) (He et al., 2016). For detailed information about the datasets, refer to Appendix D.1.

**Baselines and Cases**: Our method is compared with existing similar approaches, including FedAvg (McMahan et al., 2017), FedProx (Li et al., 2020), and centralized learning. The experiments consider four distinct cases with varying data distributions across learners. In Case 1, each learner performs random sampling of uniformly informative data. In Case 2, each learner contains different types of labels, indicating heterogeneous data across learners. In Case 3, each learner possesses the complete dataset. In Case 4, the first half of the learners contain only data samples from Case 1, while the second half contains only data samples from Case 2.

**Implementation Details**: In the simulation, we configure the number of local learners to vary between 5 and 100, with learners uniformly sampled from the dataset. We assume that the resource consumed is time, with a total time resource of 60 seconds unless otherwise specified. For more information regarding training and control parameters, please refer to Appendix D.1. Experiments are implemented using TensorFlow. Locally, experiments are conducted on CPU machines equipped with a 2.3 GHz Intel Core i7 processor and an NVIDIA 3070Ti GPU. For more resource-intensive tasks, we use a remote server equipped with a 16 vCPU Intel(R) Xeon(R) Gold 6430 processor, 120 GB of memory, and two RTX 4090 GPUs along with six RTX 2080Ti GPUs.

## 5.2 SIMULATION RESULTS

**Performance Analysis for Methods and Cases**: Figure 1 displays a comparison of the FedADM method's performance on prediction accuracy and loss over homogeneous and heterogeneous data against other baselines such as FedAvg, FedProx, and centralized learning (dataset: MNIST, classifier: SVM). It is evident that the proposed FedADM method demonstrates significant advantages regardless of the data homogeneity. This underscores the effectiveness of the proposed FedADM method and illustrates the benefits of exploiting parameter and loss dissimilarities.

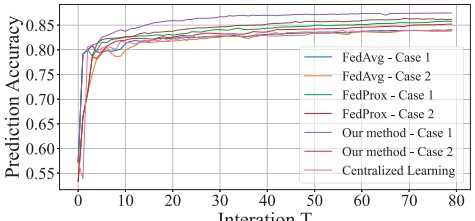 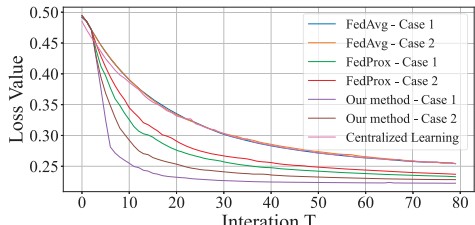

Figure 1: Comparison of prediction accuracy (left) and loss value (right) across different methods.

Figure 2 shows the impact of the optimal local update times $\tau^*$ on prediction accuracy and loss across four different cases (datasets: Fashion-MNIST, classifier: SVM). The optimal $\tau^*$ varies by case, highlighting the importance of precisely optimizing the adaptive local update times. Please refer to Appendix D.2, Figure 6 for another experiment involving the MNIST dataset and SVM classifier.

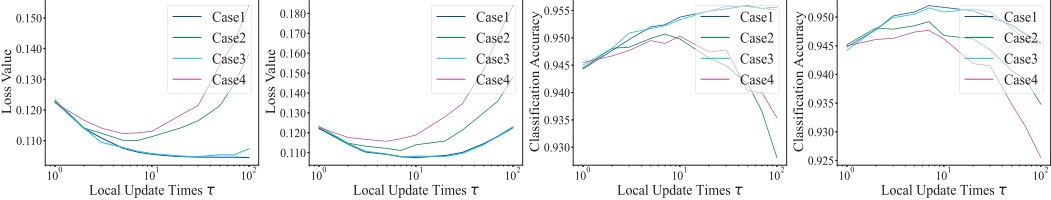

Figure 2: Impact of the optimal local update times $\tau^*$: (1) Loss on training data using FedADM, (2) Loss on training data using FedProx, (3) Prediction accuracy on testing data using FedADM, and (4) Prediction accuracy on testing data using FedProx.

**Performance Analysis under Varied Conditions**: Figure 3 presents the performance of FedADM compared to FedProx in terms of prediction accuracy and loss with varying numbers of local learners (dataset: MNIST, classifier: SVM). Considering four different cases, the results indicate that as the number of learners increases, FedADM maintains higher accuracy and lower loss. The FedADM method performs better in homogeneous data compared to heterogeneous data.

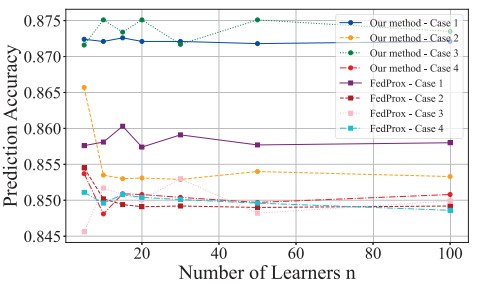 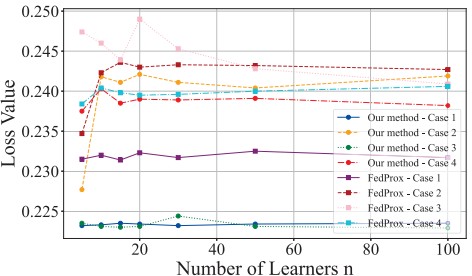

Figure 3: Comparison of the performance between the proposed FedADM and FedProx across different numbers of learners $n$ (5, 10, 15, 20, 30, 50, and 100).

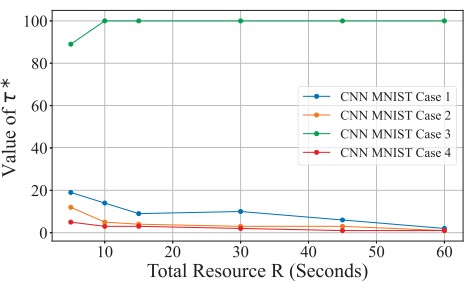 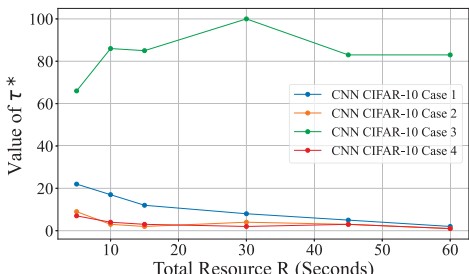

Figure 4: Comparison of $\tau^*$ values over different resource times $R$.

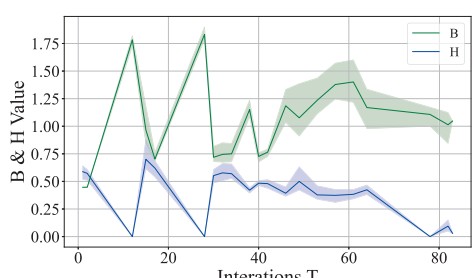 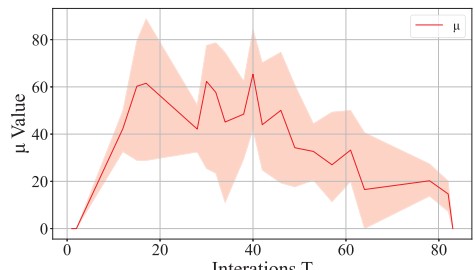

Figure 5: The behavior of $B$, $H$, and $\mu$ values over iterations $T$.

Figure 4 illustrates local update times of FedADM under different resource budgets for four cases (dataset: MNIST and CIFAR-10, classifier: CNN). The $\tau^*$ represents the local update times. As we have seen, Case 3 in both datasets shows a constant high value, indicating stable performance over time. However, for the other cases, the value of $\tau^*$ decreases significantly as the total resource increases, reflecting reduced local updates as more resources are allocated. Figure 7 shows the performance comparison of different methods as a function of total resource $R$ (datasets: MNIST, classifier: CNN). It highlights that our method consistently outperforms the other methods across different resource levels, achieving lower loss and higher accuracy. The centralized learning model shows a strong performance, but with more iterations, our method provides superior accuracy while maintaining competitive loss reduction. This figure is in Appendix D.2. Figure 8 showcases the performance in terms of prediction accuracy and loss under different datasets (dataset: MNIST and CIFAR-10, classifier: CNN). The results highlight the robustness of our method in achieving low loss and high accuracy on both datasets. This figure is in Appendix D.2.

**Convergence Behavior**: Figure 5 shows the behaviors of parameters for $\mu$ of *parameter dissimilarity* and $B_k$, $H_k$ of *loss dissimilarity*, highlighting the variations and overall trends during the process. As the iteration number $T$ increases, the proximal term coefficient $\mu$ for measuring parameter dissimilarity tends to be smaller, and the parameters $B_k$, and $H_k$ for local loss dissimilarity in the FedADM method, respectively converge to 1 and 0.

## 6 CONCLUSION

This work proposes FedADM, which utilizes *parameter dissimilarity* and *loss dissimilarity* to address data heterogeneity and reduce communication overhead in federated learning. *Parameter dissimilarity* is embedded into the local objective functions to guide local models towards approximating the global model, while *loss dissimilarity* is integrated into the surrogate function to finely control local updates and aggregation. We derive the convergence bounds for FedADM by considering the Lipschitz continuity and smoothness properties. Our experiments achieved superior performance compared to the baselines across three datasets, four cases, and two neural network models, demonstrating the convergence behavior of FedADM in resource-limited heterogeneous networks.

## 7 REPRODUCIBILITY

To set up the necessary datasets in our project, follow these steps: First, download the MNIST dataset from Yann LeCun's website and place the extracted files into the `datasets/mnist` folder in project directory. Next, for the CIFAR-10 dataset, download the CIFAR-10 binary version from Alex Krizhevsky's CIFAR page, extract the `*.bin` files, and move them to the `datasets/cifar-10-batches-bin` folder. Lastly, obtain the Fashion-MNIST dataset from the Zalando Research GitHub repository, follow the instructions for downloading, and place the dataset files into the `datasets/fashion-mnist` directory. These steps ensure that all datasets are correctly positioned for use in project.

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
