# OpenReview forum: "FedADM: Adaptive Federated Learning via Dissimilarity Measure"
_ICLR.cc/2025/Conference — ICLR 2025 Conference Withdrawn Submission_

### Official Review · Reviewer_gUAk · 2024-10-29

**Soundness:** 3
**Presentation:** 1
**Contribution:** 2
**Rating:** 3
**Confidence:** 3

**Summary:**

This paper enhances FedProx by (1) adaptively determining the optimal parameter $\mu$ that regulates the strength of the regularization term; (2) dynamically adjusting the aggregation frequency throughout the training process. The results demonstrate superior performance of FedADM compared to FedProx.

**Strengths:**

1. The motivation is clear and significant. The performance of FedProx is heavily dependent on hyper-parameters, and adjusting the aggregation frequency is crucial to consider for both performance and resource constraints.

2. The results offer intriguing insights into the trends of optimal hyper-parameters, which can inspire future research endeavors.

**Weaknesses:**

1. The writing is somewhat difficult to follow. For instance, the surrogate function is defined in Equation 23, yet the theoretical results are presented in Equation 20. Some notations are not adequately explained, such as $F^{p}$. It is challenging to comprehend the necessity of updating $B$ and $H$ from Definition 2 without reading the entire paper. Furthermore, the theoretical results are not well elucidated. It is recommended that the authors carefully refine the paper for better clarity and readability.

2. The experimental section requires improvement. For instance, providing additional results on the performance of FedProx with varying hyper-parameter values; examining the impact of different update frequencies for both FedProx and FedAvg; and exploring the integration of FedADM with more advanced federated learning algorithms to further validate its potential.

3. The updating of global models necessitates the pass of local datasets, which can be somewhat costly when the number of clients is large. It would also be beneficial to report the performance of FedADM when this process is omitted.

**Questions:**

Please see weakness.

---

### Official Review · Reviewer_a3bx · 2024-10-31

**Soundness:** 1
**Presentation:** 2
**Contribution:** 1
**Rating:** 3
**Confidence:** 3

**Summary:**

The paper presents an improvement to the FedProx algorithm by incorporating a dissimilarity-based approach, aiming to improve performance under data heterogeneity.

**Strengths:**

The work addresses an important issue in federated learning: heterogeneity in data across clients. The proposed method, using dissimilarity measures, is a novel idea to address this issue.

**Weaknesses:**

1- The method does not have client sampling, which limits its scalability for large numbers of clients.

2- The algorithm has substantial communication costs, as it requires communication with the server after each gradient computation, contrary to standard federated algorithms that allow multiple local steps to reduce communication costs.

3-  Assumption 1.2, which requires functions to be Lipschitz, is unrealistic, it doesn't hold for neural networks, especially with proxy term.  Additionally, Assumption 1.3 should properly be based on $μ_{i,k}$ instead of an aggregate between clients $\bar{\mu}_k$.

4- The parameter  $μ_{i,k}$​ is derived from the algorithm itself. How is it possible to set the learning rate in Lemma 4.2 based on its maximum values?

5- The paper lacks a direct comparison of its assumption and its bounds with state-of-the-art results and with the FedProx method.

6- Experiments are very limited. Comparison with previous works are only based on SVMs and Neural networks are only used for ablation studies.

**Questions:**

see weaknesses.

---

### Official Review · Reviewer_L1yk · 2024-11-02

**Soundness:** 3
**Presentation:** 2
**Contribution:** 2
**Rating:** 3
**Confidence:** 3

**Summary:**

This paper proposes a dissimilarity-based algorithm for solving heterogeneity challenge for FL. In particular, it incorporates two dissimilarity metrics, i.e., parameter dissimilarity and loss similarity, to regularize the local update and control aggregation, respectively. Convergence analysis under convex setting and experiments are provided.

**Strengths:**

The problem of tuning the parameter for better performance in FL is interesting and meaningful.

**Weaknesses:**

- The paper seems to be an incremental work that improves the FedProx algorithm, with refined estimation of the parameters.

- The motivation of estimating Bk Hk in Algorithm 1 is not clear, since they seem not to affect the model training.

- The loss function should be related to client $i$, like Eq 1. However, many parameters like w, K are not related to the clients, making it really confusing and hard to follow.

- The proposed method needs a lot additional computation to find the optimal parameter like linear search, then will it be more effective than the well-used vanilla grid search?

- The dissimilarity's function is not explicitly modelled. It is hard to understand how the dissimilarity affects the convergence/model training and what's the improvement based on the dissimilarity.

- The simulation setting is confusing, e.g., “the number of local learners to vary between 5 and 100"? Does the number of participating clients change over training rounds?

- There are a lot of works in FL analyzing the heterogeneous problem beyond the FedProx and FedAvg. The authors are highly suggested to add more recent baselines to show the effectiveness of the proposed heterogeneous FL algorithm.

- Why the performance of FedAMD is better than the centralized learning as centralized usually be regarded as the upper bound of FL?

- How are the effectivenesss of the parameters estimation demonstrated in the experiments? The ablation study seems shows only the parameter fine-tuning in training, but not the effectiveness of the proposed method.

**Questions:**

- For Eq 3, why the global loss is not the weighted average of local loss?

- The convergence rate is derived before the proposed algorithm. Does the proposed algorithm FedADM have the same convergence rate as Theorem 1? What’s the relationship between Theorem 1 and the proposed algorithm?

---

### Official Review · Reviewer_28Vn · 2024-11-07

**Soundness:** 1
**Presentation:** 1
**Contribution:** 2
**Rating:** 3
**Confidence:** 3

**Summary:**

The authors tackle the problem of federated learning in the presence of data heterogeneity and resource constraints. Formally, they solve the optimization problem
$$\begin{equation}
\begin{aligned}
\min_{\tau, K}\,\, &F(w) \triangleq \frac{1}{\sum_{i=1}^n D_i}\sum_{i=1}^n D_i F_i(w)\\
\text{s.t.  } & T c_m + K b_m \leq R_m, \forall m\in [M]\\
& T = K\tau
\end{aligned}\end{equation}$$
Here, $D_i$ and $F_i$ are the number of datapoints and the average loss on the $i^{th}$ client for $i\in [n]$. $K$ denotes the number of rounds, $\tau$ the number of local steps and $T$ the total number of iterations. There are $M$ resources, with $c_m$ and $b_m$ denoting the per-iteration and per-round resource consumptions and $R_m$ the resource budget for each resource $m\in [M]$. The resources can correspond to a communication, time or energy budget.

To solve this optimization problem for heterogeneous clients, the authors replace each $F_i(w)$ by the local loss function for FedProx (Li et al 2020) defined as  $F_i^p(w_i) = F_i(w_i) + \frac{\mu_{i, k}}{2}\|w_i - w_{k, global}\|^2$, where $k\in [K]$ is the round id, and $w_{k, global}$ and $\mu_{i,k}$ are the global model and regularization at that round. Then, $w_{k+1, global}$ is computed by averaging the clients' local models which are obtained after $\tau$ local steps on their local objectives. The regularization corresponds to a dual Lagrangian multiplier and is updated with an ascent step.

Under certain assumptions, some of which might not hold in practice for even simple loss functions, the authors extend the convergence analysis of (Wang et al 2019) to obtain upper bounds for $F^p$, see Theorem 1,  and in turn for $F$ in terms of $T$ and $\tau$. Following the techniques of (Wang et al 2019), the authors plug this value as an upper bound on $(1)$, and then find approximation solutions for best number of local steps $\tau^\star$.


The proposed algorithm, FedADM (Algorithm 1) performs local updates on each clients' models for a given number of local steps, after which it updates the global model, regularization, and the optimal number of local steps for next round from solving $(1)$.


**References**
- (Li et al 2020) Federated Optimization in Heterogeneous Networks. MLSys.
- (Wang et al 2019) Adaptive Federated Learning in Resource
Constrained Edge Computing Systems. JSAIT.

**Strengths:**

- **Novel combination of data heterogeneity and resource constraints**: The core idea of the paper is novel. They combine data heterogeneity from FedProx and resource constraints from (Wang et al 2019), to end up with only a per-round decision of the number of local steps to use. Further, the optimization problem they try to solve has an approximate solution found by line search.



- **Non-trivial Theory**: Although certain assumptions in theory are very strong, it is still non-trivial to combine the theoretical analysis of FedProx with changing regularization parameters with that of variable number of local steps from (Wang et al 2019).

**References**
- (Wang et al 2019) Adaptive Federated Learning in Resource Constrained Edge Computing Systems. JSAIT.

**Weaknesses:**

- **Literature review**: The authors have either omitted  references for several baselines or provided them in Section 2 but not compared their performance either theoretically or in experiments. For instance, FedLin (Mitra et al 2021) uses different local steps for each client and (Luo et al 2022) modify client selection to handle heterogeneity and client delays, a form of resource. Both these papers are baselines whose results should be compared to the proposed method as it uses variable number of local steps per round and its goal is to balance data heterogeneity and resource consumption. Further, (Reisizadeh et al 2022) also propose a method to handle delay times, and (Cho et al 2022) handle data heterogeneity via client selection. Ideally, the authors should compare to atleast some baseline using other FL techniques like client selection (Cho et al 2022, Reisizadeh et al 2022, Luo et al 2022). Without this comparison, it is unclear if selecting the number of local steps is the best technique for solving this problem.

- **Theory**:
    1. **Suboptimal analysis of FedAvg from (Wang et al 2019)** : The paper bases their algorithm and theory on the analysis of FedAvg by (Wang et al 2019). However, (Wang et al 2019) has highly suboptimal analysis of FedAvg, an example of which is that the error increases exponentially with number of local steps $\tau$ (see terms of $h(\tau)$ and $h'(\tau)$ in Theorem 1). Further, from Lemma 4.2, conditions 3) and 4) require a lower bound on the objective error, $>\epsilon$, to obtain an upper bound on the objective error (Eq 18). Additionally, the convergence rate depends on terms for which we have no control, for instance $\varphi$ which can be as small as inverse of the diameter of set of iterates, if they lie in a compact bounded set.
    2. **Better methods to handle heterogeneity**: Since (Wang et al 2019), several papers have fixed such analysis for FedAvg(Khaled et al 2020, Li et al 2020, Stich 2019) with minimal assumptions required and best possible dependence on problem parameters like polynomial dependence on number of local steps. While the authors mention some of these works in their Related Works (Section 2), they don't use their theoretical analysis. These methods use a variety of techniques like error feedback (Karimireddy et al 2020), or different local and global step sizes. Further, in several of these works, setting the step size appropriate in terms of number of local steps $\tau$ is important to change exponential dependence on $\tau$ to polynomial dependence on $\tau$. It is unclear if the theoretical  can be directly plugged into this framework yield a better algorithm and convergence rates for this resource constrained setting.
    3. **Strong assumptions**: Definitions 1, 2 and 3 are all  extremely strong heterogeneity assumptions. Some of them have been adapted from the suboptimal analysis of (Wang et al 2019). These require conditions on the iterates of the algorithm which, in general, is hard to ensure, and existing works resort to global assumptions. Even if the global versions of these definitions are satisfied, existing works require much weaker conditions to show convergence of the simpler FedAvg algorithm. The authors neither motivate nor justify the validity of these assumptions. Assumption 1 requires both Lipschitz functions and gradients, while existing works can show convergence with either. Definition 2 is especially confusing, as it uses $\ell_2$ norm on real numbers. A version of Definition 2, in terms of gradients instead of actual function values, is more intuitive and has been used previously (Yin et al 2018).

- **Algorithm and Experiments**:
    1. **Parameters used in algorithms**: The heterogeneity parameter $\xi$ in Definition 1 is required to run Algorithm 1, and the authors provide no details on how to compute it in practice. Apart from this, it seems like several quantities, like those in Definition 2 and 3, are estimated and used in Algorithm 1. This might lead to issues, for instance the denominator in $B_k$ (Eq 10) can go to $0$ by a simply subtracting a constant $F(w^\star)$ from the original objective. Additionally, all these estimates are only valid for convex problems Lipschitz and smooth problems on $\mathbb{R}^d$ with $\ell_2$ distance metric. Neural networks can be  non-smooth and due to permutation invariance of their weights $\ell_2$ norm is a poor distance metric. Therefore, these estimates are highly inaccurate for NNs and even running Algorithm 1 with these estimates might be an issue.  A minor issue is the fact that the algorithm has been derived by optimizing the theoretical result for $K$ rounds to find the number of local steps, but the in Algorithm 1, the authors recompute the number of local steps every round.

    2. **Poor Performance for CIFAR-10**: The prediction accuracy for CIFAR-10 in Figure 8 (left) is extremely low (worst is $30\%$ and best is $60\%$), even for iid client distributions, which shows that this Algorithm cannot handle non-convex models like NNs. For iid client distributions, FedAvg can achieve atleast $80\%$ accuracy (McMahan et al 2017). Also, for CIFAR-10 and MNIST with CNNs, no baselines have been plotted in Figure 8.

    3. **Limited Baselines**: As mentioned in the first weakness about literature review, the baselines are very limited, only FedProx and FedAvg. Even (Wang et al 2019), which this paper extends, has not been used as a baseline.

    4. **Limited Settings**: The authors do not test on real federated datasets with real heterogeneity (Caldas et al 2019). Further, the only resource constraint is the total time, which could be expanded to time, energy or communication, to show the flexibility of their method. Further, the authors estimate optimal number of local steps every round, without changing the resource constraints, while their algorithm allows for this.





- **Presentation** :
    1. It is unclear which of the Assumptions and Definitions are assumed and which are estimated to write Theorem 1. Definition 2 uses $\ell_2$ norm on $F$ which is a scalar.
    2. Figures 1,3,4, 7, 8 plot the performance of different heterogeneity settings in the same plot. The plots for different heterogeneity settings should be in different plots, so that we can compare the performance for each baseline for each setting.
    3. In Figure 8, the performance of two different datasets, MNIST and CIFAR10, are plotted in the same plot. These have significantly different performance, close to $99\%$ accura
    4. No description of random seeds in the experiment section or standard deviations of performance(loss or accuracy) are provided.
    5. Eq (2) should be a function of both local model $w_i$ and global model $w_{k, global}$. Then, Eq (3) is obtained by summing up Eq (2) for different local models but keeping the global model same. Eq (4) is then minimization of Eq (3) in terms of the global model. Currently, Eq (4) minimizes over global models seen in all rounds ($\{w(k\tau), \tau\in [K]\}$), which is incorrect. It should be over $\mathbb{R}^d$.
    6. Eq (8) should have $\bar{\mu}_k$ and Eq (14) should not have $\bar{\mu}_k$.
    7. Figures 1, 5 and 8 should have "iteration" instead of "iteration".
    8. Line 064: "dynamic" shoul be "dynamically".

**References**
- (Mitra et al 2021) Linear Convergence in Federated Learning: Tackling Client Heterogeneity and Sparse Gradients. NeurIPS.
- (Luo et al 2022) Tackling System and Statistical Heterogeneity for Federated Learning with Adaptive Client Sampling. INFOCOM.
- (Reisizadeh et al 2022) Straggler-Resilient Federated Learning: Leveraging the Interplay Between Statistical Accuracy and System Heterogeneity. IEEE JSAIT.
- (Cho et al 2022)  Towards Understanding Biased Client Selection in Federated Learning. AISTATS.
- (Wang et al 2019) Adaptive Federated Learning in Resource
Constrained Edge Computing Systems. JSAIT.
- (Karimireddy et al 2020) SCAFFOLD: Stochastic Controlled Averaging for Federated Learning. ICML.
- (Stich 2019) Local SGD Converges Fast and Communicates Little. ICLR.
- (Khaled et al 2020) Tighter Theory for Local SGD on Identical and Heterogeneous Data. AISTATS.
- (Li et al 2020) On the Convergence of FedAvg on Non-IID Data. ICLR.
- (Yin et al 2018) Gradient Diversity: a Key Ingredient for Scalable Distributed Learning. AISTATS.
- (Caldas et al 2019) LEAF: A Benchmark for Federated Settings. Arxiv.
- (McMahan et al 2017) Communication-Efficient Learning of Deep Networks
from Decentralized Data. AISTATS.

**Questions:**

Mentioned in weaknesses.

---

### Note · Authors · 2024-11-13

I have read and agree with the venue's withdrawal policy on behalf of myself and my co-authors.